# HugMe: Multi-view Emotion Learning on Heterogeneous Graph

## Abstract

Human emotions are the cornerstone of social interaction. Empowering machine vision with emotion perception is crucial for building harmonious and empathetic human-machine collaborative systems across a wide range of domains. Most mainstream approaches, based on a given image, adhere to the traditional image content understanding paradigm and conduct end-to-end emotion learning from the perspective of semantics-emotion association. Despite significant progress, several challenges remain. From the perspective of visual information representation, existing methods mostly adhere to the traditional image content semantics understanding paradigm and conduct end-to-end emotion learning from the perspective of content semantics-emotion semantics association, neglecting the representation and utilization of the rich structural information inherent in images. From the perspective of label information representation, existing methods mostly either directly map and classify visual features using a one-hot labeling approach, resulting in human emotion labels being treated as meaningless label indices; or they simply establish single associations between emotion labels. The heterogeneous association patterns inherent in complex human emotions have been largely unexplored. To this end, in this paper, we propose a novel **HugMe** model by **M**ulti-view **e**motion learning on **H**eterogeneo**u**s **g**raph. Specifically, for visual feature learning, we first develop a multi-view emotion representation method to leverage rich visual features from the perspectives of both semantics and structures. For label feature learning, we propose a heterogeneous emotion graph representation approach, which leverages heterogeneous graph to model the complex and diverse association patterns between different emotional labels. Finally, we develop a multi-view emotion classification module to better recognize different emotions for the given person in the image. In addition to the traditional classification loss function, to better learn and optimize our proposed HugMe, we also design a double-constraint loss function to supervise the label learning process. Extensive experiment results on well-studied human emotion benchmark datasets demonstrate the superiority and rationality of HugMe.

## 1 Introduction

Accurately identifying human emotional states (*e.g.*, happiness or sadness) from visual images is crucial for fully understanding human intentions. Due to the significant potential in many applications such as human-computer interaction, healthcare, online education, and digital entertainment, it has aroused extensive research attention in recent years Ruan et al. (2021). Different from traditional image recognition tasks (*e.g.*, object recognition, scene categorization), emotion recognition seems inherently more challenging, because it involves a much higher level of abstraction and subjectivity in the human recognition process.

Over the past few years, impressive progress has been made on human emotion learning. Early approaches analyzed emotions through faces, gestures, and other human information Fabian Benitez-Quiroz et al. (2016); Eleftheriadis et al. (2016). With the rise of context learning, more and more researchers have realized the significant role that context plays in understanding human emotions Kosti et al. (2017b). In recent years, researchers have also come to realize that a single label cannot fully capture the complexities of human emotion. To this end, some work has investigated multi-label emotion recognition Li et al. (2023), while others have attempted to learn the distribution of emo-

tion labels from continuous dimensions Xu et al. (2025). Although remarkable progress has been accomplished with these efforts, there still remain several limitations to be resolved.

**From the perspective of visual information representation**, existing methods mostly adhere to the traditional image content semantics understanding paradigm and conduct end-to-end emotion learning from the perspective of *semantics-emotion* association, neglecting the representation and utilization of the rich structural information inherent in images.

**From the perspective of label information representation**, existing methods mostly either directly map and classify visual features using a one-hot labeling approach, resulting in human emotion labels being treated as meaningless label indices; or they simply establish single associations between emotion labels. The heterogeneous association patterns inherent in complex human emotions have been largely unexplored.

To this end, in this paper, we propose a novel **HugMe** model by **M**ulti-view **e**motion learning on **H**eterogeneo**u**s **g**raph. Specifically, for visual feature learning, we first develop a multi-view emotion representation method to leverage rich visual features from the perspectives of both semantics and structures. For label feature learning, we propose a heterogeneous emotion graph representation approach, which leverages heterogeneous graph to model the complex and diverse association patterns between different emotional labels. Finally, we develop a multi-view emotion classification module to better recognize different emotions for the given person in the image. In addition to the traditional classification loss function, to better learn and optimize our proposed HugMe, we also design a double-constraint loss function to supervise the label learning process.

Our main contributions can be summarized as follows:

- We propose a MVER method, which leverages rich visual features from the perspectives of both semantics and structures.

- We develop a HEGR approach, which utilizes heterogeneous graph to model the complex and diverse association patterns between different emotional labels.

- To better optimize our proposed HugMe, we design a double-constraint loss function to supervise the label learning process in addition to the classification loss.

- Extensive experiment results on well-studied human emotion benchmark datasets demonstrate the superiority and rationality of HugMe.

## 2 RELATED WORK

Over the past few years, impressive progress has been made on human emotion learning. Early approaches analyzed emotions through action units, facial expressions, body gestures, or other human information Fabian Benitez-Quiroz et al. (2016); Eleftheriadis et al. (2016).

With the rise of context learning, more and more researchers have realized the significant role that context plays in understanding human emotions. Kosti et al. (2017a) provided a good start to infer human emotion by incorporating the context. They jointly analyzed the person and the whole scene. Lee et al. (2019) presented CAER-Net, which exploits not only human facial expression but also context information in a joint and boosting manner. Motivated by Frege's Context Principle Resnik (1967) from psychology, Mittal et al. (2020) combined three interpretations of context for emotion recognition, including multiple modalities, background context and social-dynamic context. Shen et al. (2025) exploited a multi-level context feature refinement method for emotion recognition to mitigate the impact caused by conflicting results from multi-level context.

In recent years, researchers have also come to realize that a single label cannot fully capture the complexities of human emotion. To this end, some work has investigated multi-label emotion recognition. Ruan et al. (2020) modeled the inner connections among emotion labels based on an encoder-decoder framework, and treated the multi-label classification task as a sequence generation problem. Inspired by the progress of Graph Convolutional Networks (GCN) in multi-label object detection tasks Chen et al. (2019), Li et al. (2023) proposed to model the emotion label dependency in real-world images with an emotion graph. Wu et al. (2024) studied hierarchical emotion recognition with scene graphs.

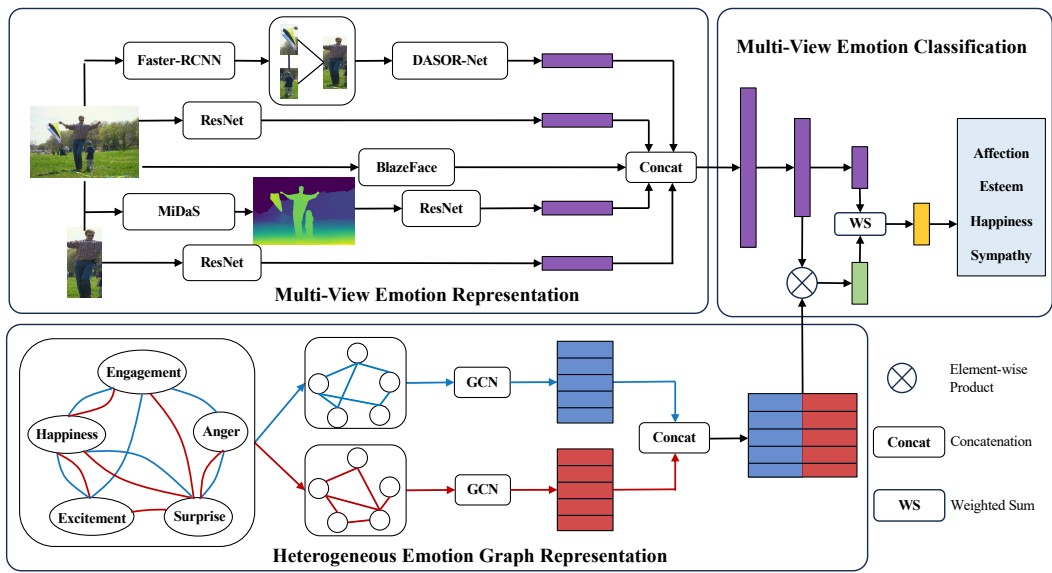

Figure 1: Overall architecture of our proposed HugMe.

Others have attempted to learn the distribution of emotion labels from continuous dimensions. Yang et al. (2021) proposed a well-grounded circular-structured representation to utilize the prior knowledge for visual emotion distribution learning. Further, Yang et al. (2022) developed a novel subjectivity appraise-and-match network (SAMNet) to investigate the subjectivity in visual emotion distribution. From the perspective of information filtering, Xu et al. (2025) proposed a multiple feature refining network to minimize low-level feature redundancy and ensure the purity of emotional information in high-level features.

## 3 METHOD

The overall architecture of our proposed HugMe is shown in Figure. 1, which consists of three main components. 1) Multi-View Emotion Representation (MVER): representing visual emotion features for both subject persons and context information from different views; 2) Heterogeneous Emotion Graph Learning (HEGL): learning emotional label embedding on a heterogeneous graph by considering both label co-occurrence and semantics similarity; 3) Multi-View Emotion Classification (MVEC): recognizing emotions from different label utilization views.

### 3.1 MULTI-VIEW EMOTION REPRESENTATION

For the subject person $I_s$, we employ ResNet He et al. (2016) pre-trained on a web-derived large-scale dataset (*i.e.*, StockEmotion Wei et al. (2020)) to extract subject body features, which can be formulated as follows:

$$\boldsymbol{f_s} = \text{ResNet}(I_s). \tag{1}$$

In addition, we also extract facial (or head) features as a supplement to body features with Blaze-Face Bazarevsky et al. (2019), which is denoted as $\boldsymbol{f_h}$.

For image context, comprehensive understanding and representation of context play a vital role in human emotion recognition. In this paper, we make full utilization of context information from both semantic and structural views. Moreover, we represent visual emotion features from the views of both global image and local objects. Specifically, we first employ a pre-trained FasterRCNN Ren et al. (2015) to detect the objects around the subject in the image, which is formulated as:

$$\boldsymbol{l}, \boldsymbol{b} = \text{FasterRCNN}(I_{mask}), \tag{2}$$

where $I_{mask}$ represents the subject-masked image. $\boldsymbol{l} \in \mathcal{R}^k$ denotes the detected $k$ object labels. $\boldsymbol{b} \in \mathcal{R}^{k \times 4}$ means coordinates of $k$ detected object bounding box.

As stated above, both semantic and structural information are important for emotion analysis. Therefore, we propose a Distance-Aware Relation Network (DARN) to further represent features of local objects. For object semantics, we consider both visual features and category features, which are represented as follows:

$$\boldsymbol{F}_o = [\text{ResNet}(I, \boldsymbol{b}); emb(\boldsymbol{l})]. \tag{3}$$

Here, we use $\text{ResNet}(I, \boldsymbol{b})$ to denote the visual feature matrix of detected object regions, and $emb(\boldsymbol{l})$ to represent the object label embedding. $\boldsymbol{F}_o$ is the extracted object semantic features. The subject-object distance information is computed as follows:

$$dist(\boldsymbol{b}_s, \boldsymbol{b}_o) = \sqrt{(b_s[0] - b_o[0])^2 + (b_s[1] - b_o[1])^2},$$
$$\boldsymbol{\alpha} = softmax(\frac{1}{dist(\boldsymbol{b}_s, \boldsymbol{b}_o)}), \boldsymbol{f}_o = \sum_{i=1}^{k} \boldsymbol{F}_{oi} \cdot \boldsymbol{\alpha}_i. \tag{4}$$

Here, $\boldsymbol{f}_o$ is the distance-aware local object feature. $\boldsymbol{\alpha} \in \mathcal{R}^k$ is the calculated relative importance weight of different objects based on distance perception. $dist(\boldsymbol{b}_s, \boldsymbol{b}_o) \in \mathcal{R}^k$ denotes the distance between the subject person and each object around.

For the global context, we employ MiDasRanftl et al. (2020) and ResNet to extract structural depth information and semantic features of the whole image, which can be formulated as:

$$\boldsymbol{f}_d = \text{MiDas}(I), \boldsymbol{f}_g = \text{ResNet}(I). \tag{5}$$

Then, we concatenate the above context features and subject person features to comprehensively represent the image emotion features, which are represented as follows:

$$\boldsymbol{f}_I = concat(\boldsymbol{f}_s; \boldsymbol{f}_h; \boldsymbol{f}_o; \boldsymbol{f}_d; \boldsymbol{f}_g). \tag{6}$$

## 3.2 HETEROGENEOUS EMOTION GRAPH LEARNING

As stated above, there obviously exist correlations among different emotional labels. For example, *'happiness'* usually has a larger probability to co-occur with *'affection'* than *'anger'*. Therefore, how to model such label dependencies and learn label representations to boost the performance of emotion recognition is a very critical challenge in our work. In this paper, we propose a Heterogeneous Emotion Graph Learning (HEGL) method to model label dependencies.

To be specific, for graph nodes, emotional labels are viewed as different nodes in the graph. We adopt the definition of emotional labels described in Kosti et al. (2017a), and employ a pretrained BERT Kenton & Toutanova (2019) to extract the semantics representation of label description to initialize the label embedding, *i.e.*, graph node embedding. We denote the initialized node embedding as $\boldsymbol{H}^{(0)}$. In this way, richer emotional semantics of labels can be obtained.

For graph edges, as shown in Equation (7), we need to construct the correlation matrix $\boldsymbol{A}$. In the emotional label graph, there are two different types of edges. Therefore, the emotional label graph is actually a heterogeneous graph. For one type, we compute the cosine similarities among label embeddings to generate the correlation matrix. For the other type, we also benefit from prior learning. As suggested by Chen et al. (2019), we model the label correlations with conditional probability. We use $P(L_j|L_i)$ to denote the probability of occurrence of $L_j$ when $L_i$ appears. To be specific, we count the occurrence of label pairs in the training set and get the matrix as $\boldsymbol{O} \in \mathbb{R}^{N \times N}$, where $O_{ij}$ means the co-occurrence times of label $L_i$ and label $L_j$. $N$ is the emotional class number. Then, we also count the occurrence times for each label in the training set, and get the vector as $\boldsymbol{m} \in \mathbb{R}^N$. The correlation matrices are computed as follows:

$$A_{ij}^h = Relu[cos(\boldsymbol{H}_i^{(0)}, \boldsymbol{H}_j^{(0)})], A_{ij}^o = O_{ij}/m_i, \tag{7}$$

where $A^h$ and $A^o$ represent the view of similarity and the view of co-occurrence, respectively. Then, emotional embedding can be updated as follows:

$$\boldsymbol{H}^{h(l+1)} = \sigma(\widehat{\boldsymbol{A}}^h \boldsymbol{H}^{h(l)} \boldsymbol{W}^{h(l)}), \widehat{\boldsymbol{A}}^h = \widetilde{\boldsymbol{D}^h}^{-\frac{1}{2}} \boldsymbol{A} \widetilde{\boldsymbol{D}^h}^{-\frac{1}{2}}, \widetilde{D}_{ii}^h = \sum_j A_{ij}^h,$$

$$\boldsymbol{H}^{o(l+1)} = \sigma(\widehat{\boldsymbol{A}}^o \boldsymbol{H}^{o(l)} \boldsymbol{W}^{o(l)}), \widehat{\boldsymbol{A}}^o = \widetilde{\boldsymbol{D}^o}^{-\frac{1}{2}} \boldsymbol{A}^o \widetilde{\boldsymbol{D}^o}^{-\frac{1}{2}}, \widetilde{D}_{ii}^o = \sum_j A_{ij}^o, \tag{8}$$

where $\boldsymbol{W}^{(l)}$ is a transformation matrix to be learned. $\widehat{\boldsymbol{A}}$ is the normalized version of the correlation matrix $\boldsymbol{A}$. $\sigma(\cdot)$ stands for non-linear activation function, i.e., $Relu$.

To incorporate them to make the correlation information more comprehensive, we concatenate the learned label embeddings as follows:

$$\boldsymbol{H} = concat(\boldsymbol{H}^{h(l+1)}; \boldsymbol{H}^{o(l+1)}). \tag{9}$$

### 3.3 MULTI-VIEW EMOTION CLASSIFICATION

For multi-class or multi-label classification work, previous methods generally have two processing methods. One is to treat each label as a meaningless one-hot vector form, and they directly map the image features to the probability distribution of the label category, which lacks the use of the rich semantic information contained in the label, especially label dependencies; the other is treating labels as embedding vectors with semantic information, they directly perform label classification by performing a dot product operation on image features and label embeddings to find similarity. However, this may introduce uncontrollable noise information due to insufficient label embedding learning. To this end, in this paper, we develop a Multi-View Emotion Classification (MVEC) method to make full utilization of label information for better emotion recognition from the two views.

Specifically, as shown in Figure 1, for the first view, we employ a multi-layer perceptron (MLP) to calculate the distribution probability over all emotional labels. Each MLP has two hidden layers with $Relu$ activation and a $softmax$ output layer, which can be formulated as follows:

$$\boldsymbol{p}_1 = \text{MLP}(\boldsymbol{f}_I), \tag{10}$$

where $\boldsymbol{p}_1 \in \mathcal{R}^n$. For the other view, we use a project layer to transform the image feature into the same space with the label embedding space, then compute the dot product between image features and each label embedding. The process can be represented as follows:

$$\boldsymbol{f}_I^{'} = \text{fc}(\boldsymbol{f}_I), \boldsymbol{p}_2 = \boldsymbol{f}_I^{'} \boldsymbol{H}. \tag{11}$$

Then, we fuse the results from two different views by weighted sum as follows:

$$\boldsymbol{p} = \beta \boldsymbol{p}_1 + (1 - \beta) \boldsymbol{p}_2, \tag{12}$$

where $\boldsymbol{p}$ is the predicted probability across all emotional labels. $\beta \in [0, 1]$ is a hyperparameter.

### 3.4 OBJECTIVE FUNCTION

We carefully design the loss function in this paper to better learn and optimize our proposed HugMe model. To start with common practice, we adopt cross-entropy loss for the classification task as $L_C$.

In HEGR module, we learn better label embeddings by modeling label dependencies. However, the learning process does not include explicit labels, which may introduce uncertainty noise. Therefore, we propose a double-constraint loss function to supervise the label-learning process, which is formulated as follows:

$$S_{ij}^o = \frac{H_i^{o(l+1)} \cdot H_j^{o(l+1)})}{||H_i^{o(l+1)}||_2 \cdot ||H_j^{o(l+1)}||_2}, S_{ij}^h = \frac{H_i^{h(l+1)} \cdot H_j^{h(l+1)}}{||H_i^{h(l+1)}||_2 \cdot ||H_j^{h(l+1)}||_2},$$

$$L_{sem} = \frac{1}{N^2} \sum_{i=1}^N \sum_{j=1}^N (S_{ij}^h - A_{ij}^h)^2, L_{occur} = \frac{1}{N^2} \sum_{i=1}^N \sum_{j=1}^N (S_{ij}^o - A_{ij}^o)^2, \tag{13}$$

$$L_{DC} = \frac{1}{2}(L_{sem} + L_{occur}),$$

where $||\cdot||$ is the $L_2$ norm. $L_{sem}$ and $L_{occur}$ are computed label constraint loss from the perspective of semantics and cooccurrence, respectively. Then, we average them to obtain the double-constraint label learning loss $L_{DC}$. The final objective function of the whole model learning is composed of the aforementioned terms, which can be formulated as follows:

$$L = \lambda L_C + (1 - \lambda)L_{DC}, \qquad (14)$$

where $\lambda$ is a hyperparameter ranging from $[0, 1]$ to balance the weight of label classification loss and label learning loss.

## 4 EXPERIMENT

In this section, we first introduce the experiment setup. Then, we evaluate the model performance on public available dataset. Next, we give detailed analyses and discussions of the model and experimental results.

### 4.1 EXPERIMENT SETUP

**Dataset.** To demonstrate the effectiveness of our proposed method, we conduct extensive experiments on the EMOTIC Kosti et al. (2017a) and CAER-S Lee et al. (2019) datasets, following previous work Zhang et al. (2019); Wu et al. (2024). EMOTIC is a collection of images of people in unconstrained environments annotated according to their apparent emotional states. It has a total size of 23,517 images annotated with 26 categories, some of which were manually collected from Web. Others are from COCO Lin et al. (2014) and Ade20k Zhou et al. (2019). CEAR-S has a total size of $70k$ images, which are obtained from TV shows. Every image in CAER-S was annotated by the annotators with a consistent label from seven basic emotion categories, including Anger, Disgust, Fear, Happy, Sad, Surprise, and Neutral. We follow the data split as the original official datasets.

**Evaluation Metrics and Baselines.** Following Wu et al. (2024), we adopt Average Precision (*AP*, area under the Precision-Recall curves) and mean AP (*mAP*) scores as evaluation metrics on EMOTIC dataset. And use *Accuracy* score for evaluation on CAER-S dataset. For a comprehensive comparison, we adopt both small models designed for emotion learning, *e.g.*, Kosti et al. (2017a); Zhang et al. (2019); Li et al. (2023); Wu et al. (2024) and multi-modal large language models (MLLM), *e.g.*, BLIP2-6.7b, LLaVa-1.6-7b, GPT-4o and Qwen2.5VL-7B-Instruct as baselines.

**Implementation Details.** For model setting, the whole image is resized to $224 \times 224$, while the head, body, and object region are resized to $128 \times 128$. The dimensions of visual embedding with ResNet-50, object label embedding, initial emotional label embedding, and emotional label embedding learned from GNN are set as $(2048, 300, 300, 512)$, respectively. $\lambda$ and $\beta$ are finally set as $0.8$ and $0.5$, respectively. For training setting, we use AdamW as optimizer with the initial learning rate of $1e - 4$, and weight decay of $2.5e - 4$ to mitigate model overfitting. Maximum training epochs on EMOTIC and CAER-S are set as 50 and 140, respectively. The batch size is set as 60 for both datasets. During training, we adopt an early stopping strategy and the Optuna tool to help train the best model.

### 4.2 OVERALL PERFORMANCE

In this subsection, we evaluate different models on EMOTIC and CAER-S datasets. As illustrated in Table 1 and Table 2, our proposed HugMe model achieves the best performance compared with baseline methods.

Among these baselines, Kosti et al. (2017b) and Zhang et al. (2019) make efforts on improving visual feature representation, while overlook the inherent dependencies among different emotional labels. To this end, Li et al. (2023) propose to model label relations through an emotion graph, and Wu et al. (2024) introduce a scene graph for further enhancement. Despite the progress, these methods still have limitations in mining heterogeneous association patterns (*e.g.*, similarity and cooccurrence) among emotional labels.

Table 1: Overall Performance (AP Scores, %) on EMOTIC Dataset.

| Categories | Kosti et al. (2017b) | Zhang et al. (2019) | Li et al. (2023) | Wu et al. (2024) | BLIP2-6.7b | LLaVa-1.6-7b | GPT-4o | QWEN-7b | Ours |
|---|---|---|---|---|---|---|---|---|---|
| Affection | 11.29 | 46.89 | 18.00 | 32.25 | 24.25 | 34.32 | 22.53 | 16.58 | 25.37 |
| Anger | 26.01 | 10.87 | 31.52 | 10.37 | 8.30 | 4.98 | 10.32 | 32.23 | 39.21 |
| Annoyance | 16.39 | 11.23 | 18.53 | 12.84 | 8.99 | 12.37 | 9.98 | 7.13 | 22.04 |
| Anticipation | 58.99 | 62.64 | 57.30 | 56.83 | 91.35 | 91.87 | 54.16 | 54.82 | 58.08 |
| Aversion | 9.56 | 5.93 | 6.60 | 7.70 | 6.45 | 6.46 | 3.51 | 5.21 | 9.42 |
| Confidence | 81.09 | 72.49 | 76.20 | 77.51 | 59.37 | 62.05 | 68.09 | 54.47 | 77.41 |
| Disapproval | 16.28 | 11.28 | 16.80 | 12.47 | 7.56 | 10.38 | 8.64 | 7.92 | 20.53 |
| Disconnection | 21.25 | 26.91 | 27.48 | 26.41 | 25.85 | 31.05 | 17.03 | 21.73 | 26.91 |
| Disquietement | 20.13 | 16.94 | 20.87 | 16.75 | 13.53 | 14.45 | 17.17 | 16.35 | 21.35 |
| Doubt | 33.57 | 18.68 | 21.04 | 18.62 | 18.58 | 19.06 | 31.20 | 17.83 | 21.48 |
| Embarrassment | 3.08 | 1.94 | 2.03 | 2.80 | 5.45 | 5.44 | 2.17 | 2.02 | 2.56 |
| Engagement | 86.27 | 88.56 | 86.14 | 86.83 | 95.14 | 96.10 | 80.77 | 79.51 | 88.05 |
| Esteem | 18.58 | 13.33 | 14.73 | 15.50 | 23.19 | 23.27 | 18.30 | 15.63 | 15.87 |
| Excitement | 78.54 | 71.89 | 70.04 | 70.73 | 64.51 | 70.36 | 69.59 | 58.37 | 71.62 |
| Fatigue | 10.31 | 13.26 | 14.36 | 12.64 | 7.68 | 11.03 | 9.53 | 13.18 | 17.08 |
| Fear | 16.44 | 4.21 | 6.49 | 5.48 | 7.84 | 8.49 | 12.79 | 6.29 | 9.97 |
| Happiness | 55.21 | 73.26 | 73.04 | 69.45 | 70.77 | 75.14 | 53.65 | 65.49 | 77.09 |
| Pain | 10.00 | 6.52 | 9.60 | 10.64 | 5.52 | 5.81 | 7.45 | 3.88 | 12.70 |
| Peace | 22.94 | 32.85 | 25.24 | 24.63 | 20.32 | 21.26 | 28.91 | 16.51 | 26.54 |
| Pleasure | 48.65 | 57.46 | 46.96 | 44.33 | 43.30 | 43.49 | 42.86 | 31.97 | 49.34 |
| Sadness | 19.29 | 25.42 | 30.80 | 21.21 | 7.06 | 18.72 | 10.70 | 16.66 | 36.69 |
| Sensitivity | 8.94 | 5.99 | 10.01 | 5.89 | 4.77 | 4.75 | 3.96 | 2.98 | 13.95 |
| Suffering | 17.60 | 23.39 | 29.88 | 21.48 | 5.22 | 6.17 | 9.74 | 9.48 | 36.72 |
| Surprise | 21.96 | 9.02 | 7.92 | 7.94 | 15.42 | 16.14 | 17.66 | 7.36 | 12.05 |
| Sympathy | 15.25 | 17.53 | 14.44 | 13.24 | 21.48 | 21.62 | 9.10 | 12.48 | 17.09 |
| Yearning | 9.01 | 10.55 | 8.78 | 9.12 | 13.05 | 12.79 | 7.93 | 6.89 | 9.15 |
| mAP | 28.33 | 28.42 | 28.64 | 26.68 | 25.96 | 27.99 | 23.80 | 22.42 | **31.47** |

Table 2: Overall Performance on CAER-S Dataset.

| Model | Accuracy | Model | Accuracy |
|---|---|---|---|
| Zhao et al. (2021b) | 85.87% | BLIP2-6.7b | 14.21% |
| Zhao et al. (2021a) | 88.42% | LLaVa-1.6-7b | 28.59% |
| Li et al. (2023) | 84.42% | GPT-4o | 29.33% |
| Wu et al. (2024) | 90.83% | QWEN-7b | 22.42% |
| **Ours** | | | **91.11%** |

The superiority of our proposed HugMe mainly lies in two aspects. For visual representation, we propose a MVER method to leverages rich visual features from the perspectives of both semantics and structures. For label representation, we develop a HEGR approach, which utilizes heterogeneous graph to model the complex and diverse association patterns between different emotional labels. As shown in Table 1 and Table 2, our proposed HugMe also outperforms MM-LLM based methods, which again demonstrate the effectiveness and superiority of HugMe.

## 4.3 ABLATION PERFORMANCE

The overall experimental results have already proven the superiority of our proposed HugMe method. However, which component is really important for performance improvement is still unclear. Thus, in this subsection, we conduct an ablation study on HugMe to examine the effectiveness of each component. The results are illustrated in Table 3. From the view of information utilization, it is obvious that the performance significantly decreased when removing any information from HugMe separately, which means these information are critical for human emotion learning. From the view of model design, when removing DARN and HEGR separately, experiment results in Table 3 again show the rationality of our proposed method.

Table 3: Ablation Performance (AP Scores, %) of HugMe on EMOTIC Dataset. Here, w/o means without.

| Categories | w/o object | w/o context | w/o head | w/o depth | w/o body | w/o cooc-curence | w/o similar-ity | w/o DARN | w/o HEGR | **HugMe** |
|---|---|---|---|---|---|---|---|---|---|---|
| **Affection** | 24.74 | 19.46 | 24.82 | 23.41 | 24.46 | 20.89 | 18.15 | 22.03 | 23.85 | 25.37 |
| **Anger** | 38.84 | 36.01 | 40.44 | 38.95 | 38.77 | 38.77 | 37.03 | 40.27 | 38.53 | 39.21 |
| **Annoyance** | 21.86 | 19.92 | 22.26 | 21.32 | 21.78 | 22.25 | 21.82 | 21.14 | 21.06 | 22.04 |
| **Anticipation** | 57.4 | 55.69 | 57.13 | 57.29 | 56.75 | 57.5 | 57.4 | 56.94 | 56.93 | 58.08 |
| **Aversion** | 9.2 | 8.12 | 9.02 | 9.15 | 8.82 | 9.24 | 9.51 | 9.36 | 8.41 | 9.42 |
| **Confidence** | 76.61 | 73.09 | 76.29 | 76.85 | 76.11 | 76.85 | 76.22 | 76.87 | 76.12 | 77.41 |
| **Disapproval** | 20.24 | 18.77 | 20.11 | 20.28 | 19.7 | 20.52 | 19.38 | 20.18 | 18.56 | 20.53 |
| **Disconnection** | 25.81 | 24.24 | 25.5 | 25.39 | 25.24 | 26.23 | 25.88 | 24.67 | 26.7 | 26.91 |
| **Disquietment** | 20.87 | 19.68 | 21.85 | 19.28 | 20.11 | 20.68 | 21.32 | 21.19 | 18.95 | 21.35 |
| **Doubt** | 21.31 | 20.56 | 21.65 | 21.29 | 20.84 | 20.95 | 22.06 | 20.82 | 19.94 | 21.48 |
| **Embarrassment** | 2.27 | 2.77 | 2.4 | 2.18 | 2.12 | 2.26 | 2.53 | 2.51 | 2.42 | 2.56 |
| **Engagement** | 87.1 | 86.25 | 86.68 | 86.37 | 87.03 | 87.28 | 87.44 | 87.05 | 86.77 | 88.05 |
| **Esteem** | 15.6 | 15.66 | 15.48 | 16.47 | 16.17 | 15.88 | 15.1 | 15.45 | 15.62 | 15.87 |
| **Excitement** | 70.52 | 68.44 | 70.53 | 70.09 | 69.48 | 70.39 | 69.77 | 70.57 | 70.61 | 71.62 |
| **Fatigue** | 17.47 | 15.56 | 16.34 | 17.66 | 17.61 | 16.93 | 15.54 | 16.28 | 17.49 | 17.08 |
| **Fear** | 10.28 | 7.64 | 9.13 | 8.61 | 9.16 | 7.89 | 7.69 | 8.39 | 7.75 | 9.97 |
| **Happiness** | 74.68 | 74.75 | 75.81 | 74.48 | 73.41 | 76.38 | 78.29 | 77.21 | 75.46 | 77.09 |
| **Pain** | 12.01 | 12.23 | 12.18 | 12.6 | 11.98 | 11.54 | 9.64 | 12.56 | 10.8 | 12.70 |
| **Peace** | 26.65 | 23.47 | 26.95 | 26.85 | 26.45 | 25.92 | 23.84 | 26.75 | 25 | 26.54 |
| **Pleasure** | 48.85 | 48.48 | 48.5 | 48.06 | 47.7 | 49.38 | 50.53 | 49.73 | 47.5 | 49.34 |
| **Sadness** | 34.09 | 30.67 | 37.05 | 33.26 | 34.35 | 34.35 | 35.57 | 35.9 | 33.51 | 36.69 |
| **Sensitivity** | 12.28 | 12.02 | 11.8 | 11.75 | 13.17 | 12.52 | 11.49 | 12.6 | 11.58 | 13.95 |
| **Suffering** | 35.04 | 31.82 | 34.5 | 31.88 | 34.3 | 33.88 | 36.39 | 36.49 | 34.23 | 36.72 |
| **Surprise** | 13.11 | 11.12 | 13.32 | 11.72 | 11.39 | 10.51 | 10.29 | 10.94 | 10.82 | 12.05 |
| **Sympathy** | 17.55 | 15.22 | 15.54 | 15.94 | 16.81 | 16.22 | 15.45 | 14.31 | 16.79 | 17.07 |
| **Yearning** | 9.65 | 9.03 | 8.78 | 9.54 | 9.54 | 8.96 | 8.66 | 8.97 | 8.33 | 9.15 |
| **mAP** | 30.92 | 29.26 | 30.93 | 30.41 | 30.48 | 30.55 | 30.27 | 30.74 | 30.14 | 31.47 |

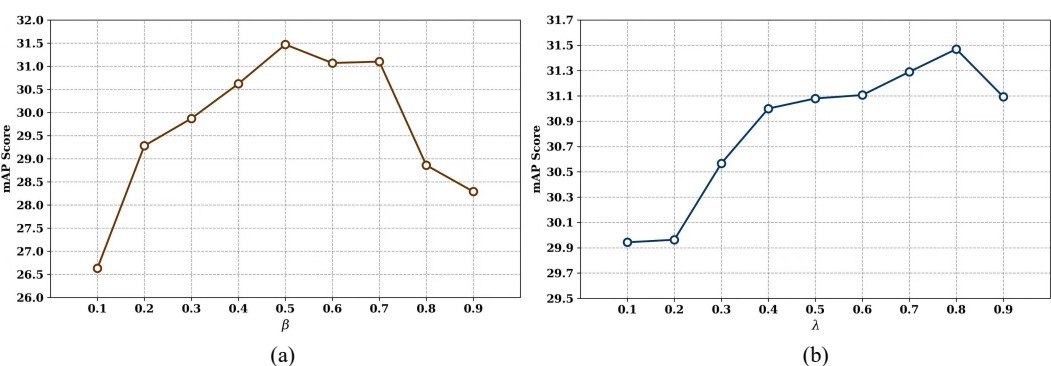

(a)  (b)

Figure 2: Parameter sensitivity study of HugMe on EMOTIC dataset with different settings of $\beta$ and $\lambda$.

## 4.4 SENSITIVITY ANALYSIS

As mentioned before, the parameter $\beta$ and $\lambda$ control the fusion manner of the final prediction and loss function, respectively. We intend to figure out how these parameters affect model performance. Thus, we conduct parameter sensitivity experiments in this subsection. Figure 2 illustrates the corresponding results.

From the experiment results, we can observe a similar varying pattern of model performance with the change of $\beta$ and $\lambda$, which demonstrates the importance of our proposed MVEC module and loss function. Overall, the trends of experimental performance change are slight, which demonstrates the robustness of our proposed HugMe.

## 5 CONCLUSION

Despite the significant progress of existing methods in human emotion recognition, there are still two main limitations unresolved from the perspectives of both visual information representation and label information representation. To this end, in this paper, we proposed a novel HugMe model by multi-view emotion learning on heterogeneous graph. Specifically, we first proposed a MVER method to leverage rich visual features from the perspectives of both semantics and structures. Moreover, we developed a HEGR approach by utilizing heterogeneous graph to model the complex and diverse association patterns between different emotional labels. To better optimize our proposed HugMe, we designed a double-constraint loss function to supervise the label learning process in addition to the classification loss. Extensive experiment results on benchmark datasets demonstrated the superiority and rationality of HugMe. Along this line, in the future, we will make efforts to combine the strengths of small models and large models in human emotion learning, from the perspective of both visual features and label correlations.

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
