# OpenReview forum: "HugMe: Multi-view Emotion Learning on Heterogeneous Graph"
_ICLR.cc/2026/Conference — ICLR 2026 Conference Withdrawn Submission_

### Official Review · Reviewer_W1sp · 2025-10-27

**Soundness:** 2
**Presentation:** 2
**Contribution:** 2
**Rating:** 2
**Confidence:** 4

**Summary:**

This paper introduces HugMe, a model for visual emotion recognition.
The authors claim existing methods fail to use structural visual information and ignore complex relationships between emotion labels.  Their model has three parts: 1) Multi-View Emotion Representation  to extract features , 2) Heterogeneous Emotion Graph Learning to model label relationships , and 3) Multi-View Emotion Classification for the final prediction.

**Strengths:**

1. The paper’s motivation to model label dependencies is strong. Using a graph to capture both semantic similarity and co-occurrence is a good idea.

2. The MVER module is comprehensive in fusing multiple feature types.

**Weaknesses:**

1. The MVER module is mostly a concatenation of features from existing pre-trained models (ResNet, BlazeFace, MiDas, etc.). It is an engineering design.

2. DARN is unclear  as the paper does not clearly explain the relationship between $F_o$ in Equation (3) and $f_o$ in Equation (5).

3. The Multi-View classification is just a simple weighted average of two standard classifiers an MLP and a dot-product head. This is a basic ensemble, not a novel fusion method.

4. The $L_{DC}$ loss seems redundant. It forces the GCN output embeddings to match the input adjacency matrices ($A^h$ and $A^o$), which likely prevents the GCN from learning any new relational information.

5. The ablation does not test the effectiveness of the two views in the MVEC module.

6. The performance gain on the CAER-S dataset is minimal, which does not strongly support the paper's claims of superiority.

**Questions:**

See weakness.

---

### Official Review · Reviewer_jkgK · 2025-10-28

**Soundness:** 2
**Presentation:** 2
**Contribution:** 2
**Rating:** 2
**Confidence:** 4

**Summary:**

This paper proposes HugMe, a multi-view emotion recognition model that leverages a heterogeneous graph to model both visual and label dependencies. The model consists of three main components: Multi-View Emotion Representation (MVER) for extracting semantic and structural visual features, Heterogeneous Emotion Graph Learning (HEGL) for capturing label correlations, and Multi-View Emotion Classification (MVEC) for combining predictions from different views. Experiments are conducted on EMOTIC and CAER-S datasets, and the authors claim superiority over several baselines, including some MLLMs.

**Strengths:**

(1) The idea of using a heterogeneous graph to model label dependencies (both co-occurrence and semantic similarity) is well-motivated and relevant.

(2) The multi-view classification approach attempts to leverage both label semantics and visual features in a unified manner.

(3) The ablation study is thorough and covers multiple components of the model.

**Weaknesses:**

(1) The core ideas (multi-view features, graph-based label modeling) have been explored in prior work. The extension to a heterogeneous graph is incremental and does not represent a significant conceptual leap.

(2) The comparison with MLLMs is not meaningful without proper fine-tuning or prompting strategies. The reported performance of MLLMs is likely underestimated.

(3) The improvements in mAP and accuracy are minimal and do not convincingly demonstrate the superiority of the proposed method.

(4) The model is evaluated only on two datasets, and the performance on certain emotion categories (e.g., Embarrassment, Fear) remains very low, suggesting limited robustness.

(5) The paper fails to explicitly define its core task. While the methodology (Section 3.2) and evaluation on EMOTIC (using mAP) clearly indicate a multi-label classification setting, this is never stated in the title, abstract, or introduction, which refer vaguely to "emotion recognition." This obscures the paper's specific contribution.

**Questions:**

(1) How does the proposed heterogeneous graph fundamentally differ from prior graph-based emotion recognition methods beyond the use of two types of edges?

(2) Why were the MLLM baselines not fine-tuned or prompted appropriately for emotion recognition? Could the authors provide a more fair comparison?

(3) The double-constraint loss is introduced to supervise label learning. How does it compare to other constraint-based or regularization methods in graph learning?

(4) The model performance on rare emotions (e.g., Embarrassment, Aversion) is very low. Does the graph structure help with imbalanced emotion recognition, and if so, how?

(5) The paper's core technical contribution is clearly tailored for multi-label emotion classification. Why is this task setting not explicitly and consistently stated from the outset? A clear declaration would significantly strengthen the motivation and context of the work.

---

### Official Review · Reviewer_ZwB7 · 2025-10-28

**Soundness:** 2
**Presentation:** 2
**Contribution:** 2
**Rating:** 2
**Confidence:** 4

**Summary:**

Existing methods overlook the rich structural information and human emotion cues embedded within images. To address this issue, this paper proposes a multi-view emotional representation approach. Extensive experiments on well-studied human emotion benchmark datasets demonstrate the superiority and effectiveness of the proposed method.

**Strengths:**

1. The research topic addressed in this paper is important. Exploring multiple types of information within images to enhance emotion recognition is a valuable and meaningful direction.

2. The methodology section is clearly described and easy to follow, and the authors have also provided code for reproducibility.

3. The proposed method achieves state-of-the-art performance.

**Weaknesses:**

1. Some descriptions in the Introduction section are unclear and difficult to follow. For example, it is not well explained what the “structural information” in the figure refers to. As far as we know, some existing methods have already considered structural information, so such a statement seems too absolute. In addition, the concept of “heterogeneous association patterns” is not clearly defined, making it difficult for us to fully understand or appreciate the authors’ research motivation.

2. Although the authors summarize existing approaches, they do not clearly articulate how the proposed method differs from prior work. Since previous studies have also employed heterogeneous GNNs for label modeling and considered multi-view information, the contribution of this paper is significantly weakened.

3. We find the heterogeneous emotion graph component confusing. It is unclear how many nodes are involved in the graph learning process and whether the learned representations possess generality, which should be validated through experiments.

4. The ablation study section lacks detailed analysis and explanation. The table headers are not clearly defined, making the results difficult to interpret.

5. The paper does not include a statement regarding the usage of large language models (LLMs). It also lacks visualization analysis and specific case studies, which would have strengthened the paper’s interpretability and insight.

**Questions:**

Please refer to the above-mentioned weaknesses.

---

### Official Review · Reviewer_qeSB · 2025-11-01

**Soundness:** 3
**Presentation:** 2
**Contribution:** 2
**Rating:** 4
**Confidence:** 3

**Summary:**

This paper proposes HugMe (Multi-view Emotion Learning on Heterogeneous Graph) for image-based human emotion recognition. The authors believe that previous methods over-relies on content/semantic features and ignore structural/contextual cues, and treat emotion labels as flat one-hot IDs or with a single relation type. HugMe builds a multi-view emotion representation (MVER) that fuses subject, face, object, depth, and global context features. In addition, it builds a heterogeneous emotion graph (HEGR) with two edge types (semantic similarity via BERT and co-occurrence from data) and learns label embeddings over it, and conducts multi-view emotion classification (MVEC) that combines an MLP head and a “label-embedding dot-product” head, plus a double-constraint loss to keep learned label similarities close to the two graphs. Experiments on EMOTIC and CAER-S show improvements over existing baselines.

**Strengths:**

The motivation is clear. The visual branch is comprehensive: including subject body, face/head, objects via detection, global semantics via ResNet, and global structure via MiDaS depth. It is interesting and effective to build two label graphs: (1) semantic-similarity graph from BERT label descriptions (ReLU’d cosine), and (2) co-occurrence graph from dataset counts. GCN-style propagation are conducted on both two graphs and the two updated label embeddings are concatenated.

The experiment results are promising.

**Weaknesses:**

None of the three ingredients — multi-branch visual fusion, dual-graph label modeling, 2-view classifier + consistency loss — is by itself a significant technical novelty compared with existing multi-label vision+GCN methods. The main contribution lies in putting them together specifically for image emotion, which is not strong.

Its better to evaluate performance on more than two datasets (e.g., GroupWalk/AFEW/RAF-DB-in-context style).

Writing should be improved, e.g., “we also design a double-constraint loss function to supervise the label learning process” appears in abstract, intro, and method almost identically.

**Questions:**

1) How to handle images with missing modalities? e.g., face not detected, object detector returns 0 boxes, MiDaS fails? Do you zero-pad or drop the view?
2) Are the two GCNs shared across datasets? For CAER-S with 7 labels, did the authors re-init the graphs?
3) What is the compute cost per image vs. a plain context-aware ResNet baseline? MiDaS + Faster R-CNN can be expensive.
4) To support the heterogeneous-graph claim, its better to visualize the learned label embedding space to show that “happiness–affection–sympathy” cluster and “anger–disapproval–annoyance” cluster as intended.
5) How to evaluate MLLMs? zero-shot prompt? fixed template? single frame?

---

### Note · Authors · 2025-11-25

I have read and agree with the venue's withdrawal policy on behalf of myself and my co-authors.